# MetaReVision: Meta-Learning with Retrieval for Visually Grounded Compositional Concept Acquisition

**Guangyue Xu**
Michigan State University
xuguang3@msu.edu

**Parisa Kordjamshidi**
Michigan State University
kordjams@msu.edu

**Joyce Chai**
University of Michigan
chaijy@umich.edu

## Abstract

Humans have the ability to learn novel compositional concepts by recalling and generalizing primitive concepts acquired from past experiences. Inspired by this observation, in this paper, we propose MetaReVision, a retrieval-enhanced meta-learning model to address the visually grounded compositional concept learning problem. The proposed MetaReVision consists of a retrieval module and a meta-learning module which are designed to incorporate retrieved primitive concepts as a supporting set to meta-train vision-language models for grounded compositional concept recognition. Through meta-learning from episodes constructed by the retriever, MetaReVision learns a generic compositional representation that can be fast updated to recognize novel compositional concepts. We create *CompCOCO* and *CompFlickr* to benchmark the grounded compositional concept learning. Our experimental results show that MetaReVision outperforms other competitive baselines and the retrieval module plays an important role in this compositional learning process.

## 1 Introduction

Learning to compose from previous experience is an important integral part of human intelligence (Fodor and Pylyshyn, 1988; Biederman and Vessel, 2006). Generally, compositional learning refers to the ability to learn a set of basic primitives and generalize these primitives in a novel scenario different from training time (Kemp and Tenenbaum, 2009; Ontañón et al., 2021). It includes various learning aspects, such as systematic generalization, productivity and substitutivity (Hupkes et al., 2020). In this work, we focus on systematic generalization within the multi-modal setting and propose a multi-modal compositional problem: Grounded Compositional Concept Learning (*GCCL*). As shown in Figure 1, in the *GCCL* setting, the models are trained with primitive concepts, such as *red* and

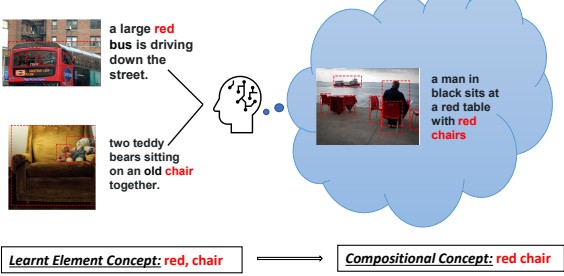

Figure 1: An illustration of Grounded Compositional Concept Learning(GCCL). For example, given concepts (red, bus) and (old, chair) in the training data, the goal is to learn to predict novel compositional concepts (red, chair) as masked token prediction at test time.

*chair*, from the training data. The trained models are then applied to predict novel compositional concepts e.g., *red chair* in the testing phase although these concepts were never seen during training.

The ideal vision-language system should have the compositional ability to solve the GCCL problem. Recently, significant efforts have been made to the development of pre-training vision-language models (VLMs) (Tan and Bansal, 2019; Su et al., 2020; Radford et al., 2021). These VLMs have demonstrated impressive performance in various downstream tasks, including Visual Question Answering (VQA) (Li et al., 2020), Vision-Language Navigation (VLN) (Hao et al., 2020) and image captioning (Zhou et al., 2020). Despite their success in related fields, it remains unclear whether these models can truly perceive the world in a compositional manner or generate language compositionally to cooperate with humans in a shared physical world. Such composition-related questions are important from both the theory and the application perspectives. From the theory perspective, compositional learning allows the model to process and understand objects by breaking them down into smaller, interpretable units. Therefore, compositional learning helps improve large models'

efficiency and generalization (Andreas et al., 2016). From the application perspective, it is not realistic to give the model all possible compositions in training data. For example, in Vision Language Navigation (VLN), it is not feasible to observe a sofa with all possible colors e.g. *red sofa and blue sofa*. The vision-language models applied in VLN are expected to recognize these compositions after learning the element concepts [1]. Compositional learning can be viewed as a special case of zero-shot learning problems. Moreover, the domain-shift problem is commonplace in zero-shot learning because the statistical distribution of the data in the training set (seen compositions) and the testing set (novel compositions) could be significantly different. While compositionality can be reliably interpreted by humans, State-of-the-art VLMs, which are trained on vast amounts of image-text pairs and employ diverse loss functions, still encounter challenges in compositional learning (Ma et al., 2023b; Thrush et al., 2022).

To address these limitations, this paper takes a closer look at the compositionality in VLM with an attempt to improve its ability. More specifically, we create two grounded compositional concept learning datasets, *CompFlickr* and *CompCOCO* curated from MSCOCO (Chen et al., 2015) and Flickr30K (Plummer et al., 2015), for VLMs' token-level compositional analysis. Moreover, we present MetaReVision, *Meta*-Learning with *Re*trieval for *Vi*sually Grounded Compositional Concept Acquisition, a retrieval-enhanced meta-learning framework for compositional concept acquisition, which introduces retriever into GCCL. The retrieval mechanism plays a crucial role in human learning. It facilitates long-term retention, understanding enhancement, and knowledge transfer during the learning process, which have been discussed by a large body of studies in cognitive science (Karpicke and Blunt, 2011; Karpicke, 2012). To mimic such human's retrieving behavior(Roediger and Butler, 2011; Karpicke and Roediger III, 2008), MetaReVision retrieves relevant primitive concepts from a pre-constructed concept database and provides them as support evidence to do meta-learning for compositional concept learning. MetaReVision follows a *Learn-Retrieval-Compose* framework. It shares the compositional learning burden between VLMs and the retriever. Through meta-

---

[1]Element concepts are also called primitive concepts in our setting. We use them interchangeably in this work.

learning from the episodes constructed by the retriever, MetaReVision learns a generalized compositional representation that can be fast updated for novel compositional recognition. We evaluate MetaReVision on the proposed CompFlickr and CompCOCO datasets. The empirical results show that coupling retrieval and meta-learning performs better in GCCL compared with previous baselines.

Contributions of this work can be summarized as follows:

- This work explores a novel angle of retrieval-enhanced compositional concept learning. The model relies on retrieval to construct episodes for meta-learning. It addresses the domain-shift problem in compositional learning by learning from the retrieved instances.

- Two datasets are created to serve as benchmarks for grounded compositional concept learning. These datasets enrich existing zero-shot vision-language tasks, from the end-task level to the token-level.

- Our experiments show that MetaReVision demonstrates stronger performance in *GCCL*, especially in the novel setting. This empirically shows the effectiveness of combining retrieval and meta-learning techniques in the context of grounded compositional learning.

## 2 Related Works

**Meta-Learning** also known as *learning to learn*, aims to solve a low-resource problem by leveraging the learned experience from a set of related tasks. Meta-learning algorithms deal with the problem of efficient learning so that they can learn new concepts or skills fast with just a few seen examples (few-shot setting) or even without seen examples (zero-shot setting). Different from the typical meta-learning scenario where the training and test episodes are given in advance in few-shot learning (Sung et al., 2018; Snell et al., 2017; Nichol et al., 2018a; Finn et al., 2017), in GCCL, we need to construct episodes to employ meta-learning methods for compositional concept learning. In MetaReVision, we introduce a retriever to actively construct episodes to help compositional concept learning. During the test time, with additional retrieved support items, MetaReVision can further fast-update VLMs for current compositional concept recognition in the query set. This test-time fine-tuning is different from previous works

which apply meta-learning in the zero-shot setting(Conklin et al., 2021).

**Retrieval-Enhanced Learning**. Retrieving related instances from a database, either the training set or external knowledge base, has been widely applied in tasks such as language modeling (Khandelwal et al., 2019), reinforcement learning (Goyal et al., 2022) and language tasks such as NER (Wang et al., 2021). Instead of distilling all training information into the model's parameters through gradient updates, retrieval-enhanced learning introduces a retriever to find related instances and based on these instances conduct further learning. For example, kNN-LM (Khandelwal et al., 2019) extends the pre-trained language model by linearly interpolating its next word distribution with a retrieval module. This design shows effective domain adaptation ability. Wang et al. finds external contexts for the target instance by retrieving a set of semantically relevant texts to fine-tune the CRF module to address the NER problem. These studies highlight the significance of actively recalling information from a database to enhance learning outcomes. The general scheme of such methods is to combine a parametric model with a non-parametric retrieval system (Long et al., 2022). Different from these settings, in *GCCL*, we train our own concept retriever and show retrieval's importance in compositional learning.

**Compositional Learning**. Recent research suggests that compositionality remains a challenge for state-of-the-art (SoTA) neural models such as Transformers and Graph Neural Networks (Nikolaus et al., 2019; Hupkes et al., 2020; SHAO et al., 2023). To tackle this challenge, inspired by symbolic AI, some works try to add structural constraints into neural models (Bergen et al., 2021). There are also some attempts to generate new data for the compositions (Naeem et al., 2023; Xian et al., 2018). Also, there have been noteworthy advancements in vision-language benchmarks that focus on probing and enhancing VLM's compositional abilities recently (Eisenschlos et al., 2023; Thrush et al., 2022; Ruis et al., 2020; Ma et al., 2023b). Nevertheless, these works build end tasks in a compositional manner. They emphasize the performance of these compositional end tasks without giving consideration to the token-level compositional ability. However, *GCCL* targets VLM's token-level compositional ability. Moreover, different from symbolic and data-augment solutions,

MetaReVision explores the retrieval method to solve the compositional problem.

# 3 Grounded Compositional Concept Learning (GCCL)

We start by introducing the settings of *Grounded Compositional Concept Learning (GCCL)* and further introduce the benchmarks we curated for this problem in this section.

## 3.1 Problem Definition

Existing VLMs try to learn a generic representation for multi-modal tokens in different contexts. These VLMs are expected to obtain generic token representations that have strong transfer ability for downstream tasks. We consider a setting that directly examines whether VLMs have the ability to acquire compositional meanings of tokens through the lens of language modeling. Different from the task-level compositional studies, *GCCL* approaches the compositional problem from the token-level and investigates whether VLMs possess the capability to acquire the compositional meanings of tokens.

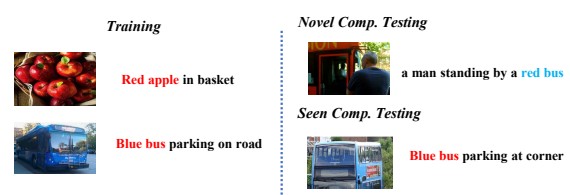

Figure 2: *GCCL* task definition. Red highlights seen compositional concepts and blue highlights novel compositional concepts.

Figure 2 shows an example of the *GCCL* task. Given a set of image-caption pairs with the compositional concepts masked out from the caption, the model is tasked to learn the concept representations and predict the masked compositional concept conditioned on the contextual information. The learned model is then applied in the testing phase on both novel compositions as well as seen compositions. The model is evaluated based on its ability to learn novel compositions while maintaining (i.e., not forgetting) seen compositions.

Formally, given a set of text-image pairs $\{(x_{cap}, x_{img})\}_{i=1}^{n}$ where $x_{img} \in \mathcal{I}$ is the image with annotated bounding boxes, $x_{cap} \in \mathcal{T}$ is the caption with the compositional concepts replaced by *MASK*. The objective of *GCCL* is to predict the masked tokens based on the contextual information(Ma et al., 2023a; Jin et al., 2020). Therefore,

for BBoxes, only the locations are considered as input, not their label information. A model capable of solving *GCCL* can be described as a functional $f : \mathcal{I} \times \mathcal{T} \rightarrow \mathcal{V}_{attr} \times \mathcal{V}_{obj}$, where $\mathcal{V}_{attr} \times \mathcal{V}_{obj}$ is the target compositional concepts which could be either *adjective + noun* pairs or *noun + verb* pairs. Based on whether $\mathcal{V}_{attr} \times \mathcal{V}_{obj}$ have been seen during training, *GCCL* can be categorized into *seen compositional testing* and *novel compositional testing*. The desired compositional VLMs should achieve improved novel performance without sacrificing the seen performance.

## 3.2 *GCCL* Dataset Creation

We build *GCCL*'s benchmarks, *CompFlickr* and *CompCOCO*, from MSCOCO (Chen et al., 2015) and Flickr30K (Plummer et al., 2015). We use the same data split introduced by Nikolaus et al.. Their work studies the composition ability of image captioning systems by selecting 24 pairs as novel compositions by removing all images related to these 24 pairs from the training dataset. This ensures that novel compositions have never been seen during training. Other works adapt the same data split for compositional learning studies. For example, Jin et al. utilized this split to check current *VL* models' compositional ability on phrases under the continual learning setting. However, in Jin et al.'s work, most of the extracted phrases are in the form of *article + noun*, like the car and a man. They are single objects instead of compositional concepts. Such phase evaluation is not a good setting for compositional learning.

In order to evaluate the token-level compositional ability, we develop two benchmarks *Compt-COCO* and *ComptFlickr* to address the above limitation. Concretely, after paring the captions using Stanta (Qi et al., 2020), we use a number of rules to collect and mask the compositional concepts, the details are in the Appendix C. Finally, the dataset is divided into 4 parts: training set without novel compositions, validation set with both seen and novel compositions for hyper-parameter tuning and model selection, seen test set, and novel test set. The detailed statistics of novel compositions for these two datasets are shown in Appendix D.

## 4 Meta-Learning with Retrieval for *GCCL* (`MetaReVision`)

`MetaReVision` mainly consists of two modules: the retrieval model and the meta-learner as shown in Figure 3. The retrieval module learns to find similar element concepts from the training data. The meta-learner organizes the retrieved items as a pseudo task to meta-tune VLMs for compositional learning. In this part, we will discuss the base VLMs, retrieval module, and meta-learning module in detail and answer two key questions in `MetaReVision`'s design: 1) How to retrieve related items, 2) How to utilize the retrieved items in the context of meta-learning.

## 4.1 Vision-Language Models (VLMs)

VLBERT (Su et al., 2020) and LXMERT (Tan and Bansal, 2019) are two representative VLMs that are suitable in our *GCCL* setting. They represent one-stream and two-stream VLMs separately. The difference is that two-stream VLMs have additional self-attention layers before cross-attention layers. We conduct experiments using these two types of VLMs to show the general effectiveness of the proposed framework. Moreover, all VLMs are trained from scratch to make sure that they do not see novel compositions during their training time.

## 4.2 Retriever and Element Concept Database

Given the compositional concepts, the ideal retriever is expected to retrieve the training examples that are the most beneficial for the target compositional concept learning. It is usually assumed that the examples that are the nearest neighbors of query examples are more likely to be beneficial ones for generalizing (Long et al., 2022). *GCCL* retriever needs an encoder to encode the element concept, construct a database to organize these element concepts' information, and retrieve relevant concepts.

**Element Concept Encoder**. Given the linguistic and visual clues for the compositional concepts, the encoder is acting as a function $f(x_{cap}, x_{img})$ that maps a *MASK* concept to a fixed-length vector $\mathbb{R}^d$. Then for each primitive concept in the target compositions, $f(\cdot)$ can help retrieve related primitive concepts. `MetaReVision` relies on these retrieved concepts to conduct further compositional learning. In this way, `MetaReVision` enhances its own compositional capability by augmenting the input through the retrieval procedure. The encoding function $f(\cdot)$ is the key component for the retriever. In traditional vision-language tasks, like VQA and Visual Entailment(Song et al., 2022), CLIP (Radford et al., 2021) is usually used as the encoder to encode the whole visual or textural input and

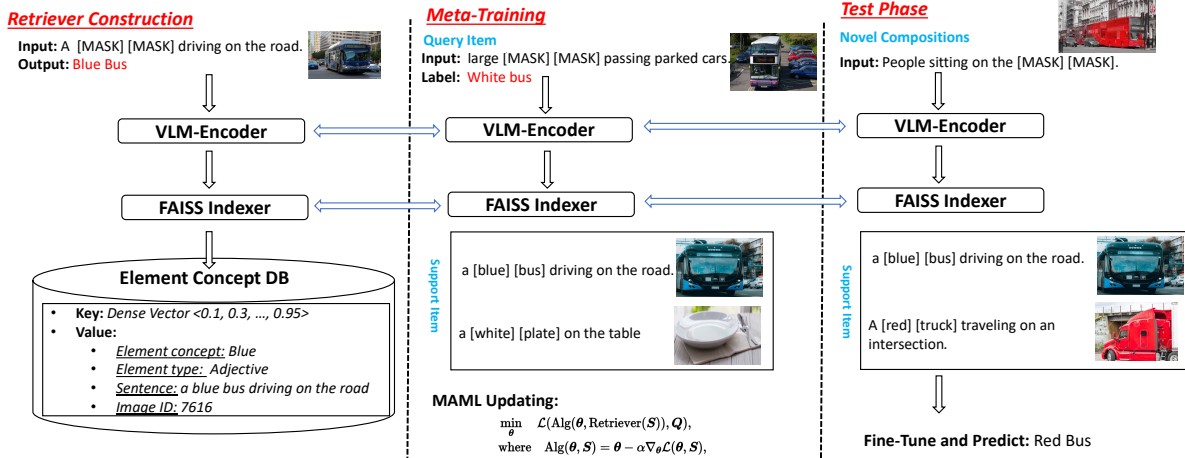

Figure 3: MetaReVision Architecture. The whole system includes two modules: retrieve and meta-trained VLM. During testing, MetaReVision retrieves related instances to fast-update VLM for novel compositional learning.

help build the retriever. However, in *GCCL*'s token-level compositional setting, we focus on the token's representation and therefore use the VLMs as an encoder to extract *MASK* concept's representation for further compositional learning. These vectors are used as keys to construct the *Element Concept Database* and perform an approximate nearest neighbor search to augment compositional learning. We add a two-layer MLP and adopt Masked Language Modeling (MLM) to train vision-language retriever. For the encoder's training, since we focus on concept acquisition, words in compositional concepts are masked with a probability of 1.0, and others are not masked during training.

**Element Concept Database**. The element concept datastore $\mathcal{DB} = \{(k_i, v_i)\}$, which is constructed offline using the above-trained vision-language encoder, consists of dense representations of masked element concepts $k = Enc(x_{cap}, x_{img}) \in \mathbb{R}^d$ is as keys and the corresponding $(x_{cap}, x_{img})$ as values. To efficiently access this database, we implement the dense retriever for *GCCL* by an off-the-shelf-retriever engine *FAISS* (Johnson et al., 2019) with a flat index (IndexFlatIP) without any training. Then given a masked concept, we can retrieve the top-K DB items by calculating the cosine similarity scores between the *[MASK]* concept with all DB items in nearly real-time as follows:

$$\text{Ret}(k) = \{(k_1, \text{Val}_1), \dots, (k_M, \text{Val}_M)\} \quad (1)$$

where $k$ is the mask concept's embedding vector, $k_i$ is the DB item's key, $\text{Val}_i = (x_{cap_i}, x_{img_i})$ is the

retrieved DB item's value, and $Ret$ is the retrieved DB item set.

After adding the retrieval module into *GCCL*, the problem can be re-formulized as:

$$p(v \mid x) = \underbrace{p(v \mid x, Ret(x))}_{Learner} \underbrace{p(Ret(x) \mid x)}_{Retrieval} \quad (2)$$

where $v$ is the *MASK* compositional concept's prediction, $x \in \mathbb{R}^d$ is the maksed concept's encoded vector and $Ret(x)$ is the retrieved DB items based on its vector $x$ as Equation 1. The compositional learning happens in two levels: 1) retrieve related items from DB based on the encoding vector, 2) learn conditioned on contextual information and the retrieved items.

### 4.3 Meta-Learning for *GCCL*

Given the retrieved items, there are several ways to exploit these examples to facilitate compositional learning. The most direct method is to fine-tuning (FT). However, because the retrieved items are noisy and FT often faces over-fitting issues when they learn from a few labeled examples, FT does not help *GCCL*. Another choice in in-context learning (Wei et al., 2022). However, as *GCCL* is a multi-modal problem. We have multiple image-caption pairs in the contextual input, current large multi-modals, like LLaVA (Liu et al., 2023) and GPT-4 (gpt, 2023), can not be applied directly here. In MetaReVision, we choose meta-learning framework to utilize the retrieved items for *GCCL*. Meta-learning here is to train the base VLM with the ability to accumulate knowledge across episodes[2]

---

[2]episodes also called tasks in meta-learning.

and build internal generic representations for tokens that are suitable for compositional learning. Moreover, we introduce the verbalizer module to enforce the predicted concept for the query set coming from the retrieved support items. The verbalizer helps mitigate the *memorization* problem in meta-learning (Yin et al., 2019). In the following part, we will discuss episode construction, the details about MAML, and *verbalizer* module used in MetaReVision.

**Episode Constructions**. We construct *GCCL* tasks $\tau_i$ for meta-learning as follows:

$$\tau_i = \left( \mathcal{D}_{\tau_i}^{\text{support}}, \mathcal{D}_{\tau_i}^{\text{query}} \right), \quad (3)$$

where $\mathcal{D}_{\tau_i}^{\text{support}}$ indicates the support set and $\mathcal{D}_{\tau_i}^{\text{query}}$ indicates the query set. Specifically, for one task, we randomly select one compositional concept as the query set. Then we retrieve a small number of examples that are similar to the query concepts. These retrieved items make up the support set. Meta-learning's objective in *GCCL* is to predict the compositional concepts in the query set after learning the element concepts in the support set. Here, episodes help VLMs to accumulate compositional knowledge and learn a generic compositional representation for masked concepts from the task-level instead of instance-level.

**Meta-Learner**. We use MAML (Finn et al., 2017) as our meta-learning algorithm. As an optimization-based method, MAML has two optimizing steps within each episode: the meta-train step and the meta-test step. In the meta-train step, MAML learns a task-specific learner $\theta'$ based on the current parameter $\theta$ and retrieved support items $S$. In the meta-test step, MAML updates the parameter $\theta$ based on the fast-updated parameter $\theta'$ and the compositional query items $Q$ as shown in Figure 4. Moreover, MAML can be solved by formulating it as a bi-level optimization problem. Equation 2 can be extended to Equation 4.

$$\min_{\boldsymbol{\theta}} \quad \mathcal{L}\left(\text{Alg}\left(\boldsymbol{\theta}, \text{Retriever}\left(\boldsymbol{S}\right)\right), \boldsymbol{Q}\right),$$
$$\text{where} \quad \text{Alg}(\boldsymbol{\theta}, \boldsymbol{S}) = \boldsymbol{\theta} - \alpha\nabla_{\boldsymbol{\theta}}\mathcal{L}(\boldsymbol{\theta}, \boldsymbol{S}), \quad (4)$$

where $\boldsymbol{\theta}$ is the learnt parameters, Retriever($\boldsymbol{S}$) stands for the retrieved DB items, $\boldsymbol{Q}$ is target compositional concept and Alg represents the optimization algorithm adapting to the support instances. There are different versions regarding Alg (Nichol et al., 2018b; Finn et al., 2017). We use MAML

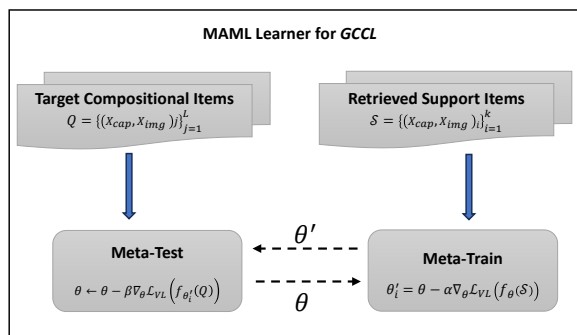

Figure 4: MAML's computing procedure

which unrolls the optimizing process and tries to find a good initial parameter configuration for all compositions.

**Verbalizer.** MAML's classical application is in few-shot learning, where class-to-label assignment needs to be conducted within each episode, that is, the same class has different labels among different episodes. Without such re-assignment, the models can memorize the class information and conduct prediction directly without considering the items in the support set. This is known as *memorization* problem in MAML discussed in (Yin et al., 2019). To help MetaReVision learn from the retrieved instances, we introduce the *verbalizer* module into MetaReVision. It enforces prediction for the query set by selecting concepts from the support set as shown in Figure 5. In this way, MetaReVision will rely on the retrieved element concepts rather than memorizing the labels to do compositional learning. This helps alleviate the MAML's memorization problem.

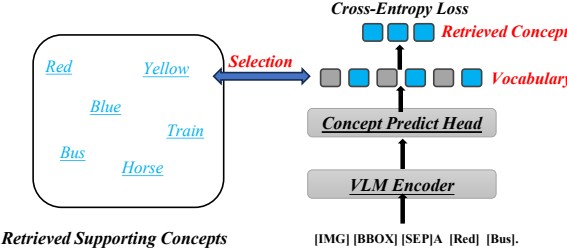

Figure 5: Verbalizer helps VLM consider retrieved instances when learning.

## 4.4 Inference

During inference time, we consider each test compositional concept as a query item and retrieve relevant instances from concept DB as support instances. Therefore, we construct a specific task for

the current compositional concept. Instead of applying the general model $\theta$ directly, MetaReVision retrieves support instances to fast-update the model to adapt to current compositions and make predictions as $v_i = \text{argmax}_{v \in Sup} P(v)$, where the prediction comes from the retrieved concepts. In MAML's testing, it is observed that a larger number of updates can give a considerable performance boost. Thus, we choose the inner loop updates to 20 before testing.

## 5 Experiments

In this section, we introduce the *GCCL*'s datasets, demonstrate the implementing details of MetaReVision, and compare its results with other baselines. Ultimately, we empirically analyze the retriever importance in MeaReVision.

### 5.1 Dataset

*CompCOCO* is constructed from MSCOCO (Chen et al., 2015) using its 2014's split. In this split, COCO-captions has $103175$ training images and $15112$ validation images (Chen et al., 2015). Because MSCOCO does not provide test data, we use the validation data as the testing data in *CompCOCO*. Moreover, we did some minor synonym modifications described in the Appendix A to extract more clean concepts.

*CompFlickr* is constructed from Flickr30k Entities (Plummer et al., 2015). Flickr30k contains $276k$ manually annotated bounding boxes for $31,783$ images and a total of $158,915$ English captions (five per image). We use the given train/val/test split to construct *CompFlickr*.

### 5.2 Evaluation Metrics.

We use accuracy as our primary metric to measure the GCCL performance and report object, attribute, and compositional accuracy separately. Jin et al. uses perplexity as the forgetting metric in continual learning which is not appropriate in our work due to MetaReVision's offline setting.

### 5.3 Implementation Details

The implementation of MetaReVision uses the HuggingFace Transformers library (Wolf et al., 2020). For MAML, we use Adam optimizer (Kingma and Ba, 2014) as both inner and outer optimizers. We set the inner learning rate to $5e-5$, the outer learning rate to $1e-5$, and based on

HIGHER [3] to calculate the higher gradients. The code for this paper will be released at [4].

### 5.4 Baselines

We use two types of baselines in this evaluation. The first is the *train-from-scratch baseline* which trains VLMs from random initialized parameters. Another baseline is *MAML without retriever*. In this setting, VLMs are meta-trained using the same retrieved tasks, but VLMs can not access the support set. It predicts directly during test time. This baseline is used to show the importance of the retriever during test time for GCCL. Moreover, we also compare two variants of MetaReVision, including *Top 4* and *Div 4*. *Top 4* retrieves top $4$ similar concepts, which may contain duplicated concepts. The same concept could have different vector representation which is affected by different visual and textual contexts. For example, car could have different vector values when modified by *red* or *blue*. *Div 4* retrieves the top $4$ distinct similar concepts expecting that the true primitive concept will be in the retrieved set.

### 5.5 Main Results

We report the performance under both novel and seen settings as shown in Table 1 and Table 2. From the two tables, we can see that MetaReVision does help compositional learning, especially in the novel setting.

**Novel Compositions**. As shown in Table 1, MetaReVision improves the performance on the novel setting compared to the pre-trained model and MAML models. This suggests that MetaReVision captures a generic representation which is beneficial for compositional learning through meta-learning on the retrieved tasks. However, compared with seen compositions (i.e., Table 2), the performance on novel pairs drops significantly across the board. MetaReVision's accuracy drops by about $20\%$ on *CompCOCO* dataset in novel setting compared with the seen setting. This indicates that such compositional generalization is still a very difficult and open task for current *VL* models.

**Seen Compositions.** Table 2 shows the performance in the seen setting. From the table, we can see that all models have similar accuracy in the seen setting. One possible reason is that all the

---

[3]https://github.com/facebookresearch/higher
[4]https://github.com/HLR/MetaReVision

| VL-Model | VLBERT | | | LXMERT | | |
|---|---|---|---|---|---|---|
| Metric | Pair Accu.↑ | Attr. Accu.↑ | Obj. Accu.↑ | Pair Accu.↑ | Attr. Accu.↑ | Obj. Accu.↑ |
| **COCO** Train-Scratch | 7.73% | 25.88% | 50.74% | 8.14% | 26.36% | 55.06% |
| MAML w/o *Ret* | 9.03% | *27.08%* | 50.04% | *9.04%* | *27.01%* | *56.19%* |
| *Ours(Top 4)* | 11.15% | 29.84% | 50.17% | 12.01% | 29.36% | 58.81% |
| *Ours(Div 4)* | 13.50% | 31.85% | 50.92% | 13.79% | 33.76% | 59.87% |
| **Flickr** Train-Scratch | 6.04% | 17.53% | 65.21% | 5.12% | 18.10% | 61.68% |
| MAML w/o *Ret* | 8.60% | 22.06% | 64.38% | 7.52% | 18.45% | 64.55% |
| *Ours(Top 4)* | 10.7% | 24.58% | 65.54% | 9.38% | 20.45% | 65.10% |
| *Ours(Div 4)* | 11.50% | 25.49% | 66.58% | 10.58% | 22.45% | 65.15% |

Table 1: Results on Novel Compositional Concept.

| VL-Model | VLBERT | | | LXMERT | | |
|---|---|---|---|---|---|---|
| Metric | Pair Accu.↑ | Attr. Accu.↑ | Obj. Accu.↑ | Pair Accu.↑ | Attr. Accu.↑ | Obj. Accu.↑ |
| **COCO** Train-Scratch | 32.45% | 49.06% | 60.03% | 34.12% | 50.33% | 61.96% |
| MAML w/o *Ret* | 32.23% | 49.05% | 59.20% | 34.09% | 49.97% | 61.93% |
| *Ours(Top 4)* | 32.27% | 49.15% | 59.98% | 34.02% | 49.90% | 61.90% |
| *Ours(Div 4)* | 32.46% | 50.01% | 60.05% | 34.15% | 50.32% | 62.00% |
| **Flickr** Train-Scratch | 24.34% | 42.72% | 52.53% | 22.68% | 40.86% | 50.11% |
| MAML w/o *Ret* | 23.73% | 41.92% | 49.01% | 22.15% | 41.21% | 49.97% |
| *Ours(Top 4)* | 23.75% | 41.95% | 49.04% | 22.75% | 41.19% | 50.01% |
| *Ours(Div 4)* | 26.52% | 46.11% | 53.23% | 23.41% | 42.02% | 51.61% |

Table 2: Results on Seen Compositional Concept.

models have been fully trained using the seen compositional concepts. MAML-based methods do not hurt the in-domain performance during this meta-learning phase.

## 5.6 Empirical Analysis of Retriever

**Retrieval Accuracy**. Figure 6 shows the retriever's top-4 accuracy for attributes, objects, and pairs under both seen and novel settings. Attribute recognition is the key challenge compared with object recognition in *GCCL*, even in the retrieval phase. In *GCCL*, the learned VLMs are biased to the seen attributes that need to be adjusted for effective compositional learning.

**Importance of diverse sampling**. Retrieving true concepts into the support set is important for *GCCL*. In this part, we assume an oracle situation where we can always select the true element concepts into the support set during test time. We study potential advantages that can be derived under this configuration. From Figure 7, we can see that the true concept in the support set does help the compositional learning. It also explains the importance of diverse sampling which increases the probability of selecting the correct elemental concepts.

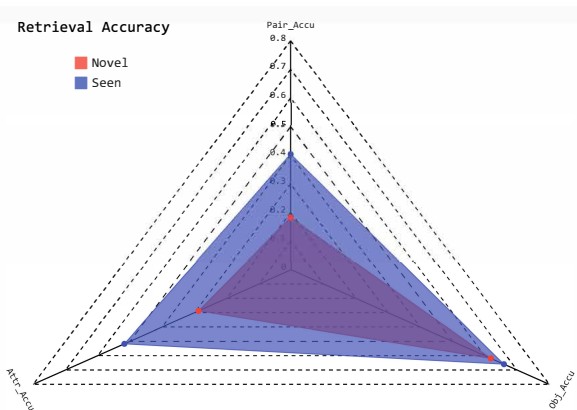

Figure 6: Retriever accuracy comparison between seen pairs and novel pairs in *CompCOCO*.

## 6 Conclusions and Future Work

In this work, we propose `MetaReVision`, which combines retrieving method and meta-learning to train VLMs for grounded compositional concept learning. Our work highlights the significance of retrieval in compositional learning. Our empirical results on two proposed datasets, *CompCOCO* and *CompFlickr*, have shown that `MetaReVision` consistently outperforms conventional VLMs and meta-learning methods without retriever, especially in novel settings. However, *GCCL* is still a chal-

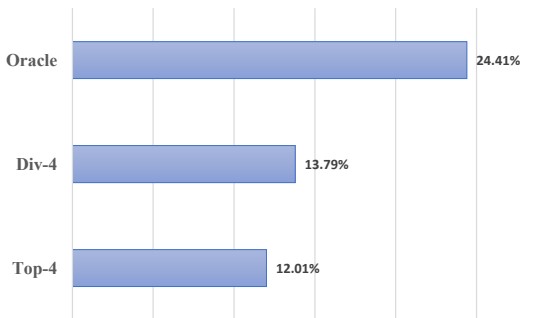

Figure 7: `MetaReVision`'s accuracy on *CompCOCO* using different retrievers.

lenging open problem and many problems remain. Our future work will explore more cognitively plausible models and explicitly address the grounding ability in compositional concept learning.

# 7 Acknowledgement

This project is supported by National Science Foundation (NSF) CAREER award 2028626 and partially supported by the Office of Naval Research (ONR) grant N00014-20-1-2005. Any opinions, findings, and conclusions or recommendations expressed in this material are those of the authors and do not necessarily reflect the views of the National Science Foundation nor the Office of Naval Research. We thank all reviewers for their thoughtful comments and suggestions.

# 8 Limitations

The limitations of the proposed `MetaReVision` include 1) Grounding limitation. Currently, we rely on VLM's attention mechanism to do grounding. We do not have an explicit grounding design to align the textual concepts and visual regions. This could be an interesting direction for future *GCCL* works. 2) SoTA generative model comparisons. Currently, we can not directly apply SoTA generative models, such as BLIP-2 and MiniGPT, on *GCCL* due to the following reasons. One reason is the *GCCL problem setting*. In GCCL, it is not easy to transform the supporting items, including multiple images and captions, into contextual input for these generative models. Another reason is *controlled evaluation* which means that these huge generative models may have already seen the novel compositions during training and it is not a fair comparison with other models. 3) Updating retriever. We construct our element concept DB in advance and not updating this DB during the meta-learning time. Training both the learner and the retriever in an end-to-end manner could improve the performance for *GCCL* and other retrieval-enhanced models.

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

## A Modified MSCOCO Synonym List

In order to extract more compositional concepts, we modify (Lu et al., 2018)'s category and change the drier synonym list as: hair drier, hairdryer, hair dryer, blow dryer, blow drier.

## B Episode Examples

Table 3 shows episode examples constructed in MetaReVision. From the table, we can see that MetaReVision can retrieve true element concepts for target compositional concepts, such as white truck, bird fly, boy eat. But there also exist cases we can not find true element concepts in the retrieved support set, such as blue bus. In this example, MetaReVision can retrieve many similar objects, but has a challenge to retrieve the true color *blue*. Also, from these randomly sampled episodes, we can see that in *GCCL*, *objects* are easier to be retrieved compared to *objects*.

| Target Context | Target Concepts | Retrieved Context | Retrieved Concepts |
|---|---|---|---|
| A white truck parked in front of a house that is being built. | White Truck | Several bikes parked next to a white van. | White Van |
| | | A man in a suit poses by an colored truck. | Colored Truck |
| | | A woman smiling in front of a big bus. | Big Bus |
| | | People waiting on the side of the road for the yellow bus. | Yellow Bus. |
| A couple of birds flying through a cloudy sky. | bird fly | Two geese are flying in the air near trees. | Geese Fly |
| | | Two hawks flying near a snow covered mountain. | Hawk Fly |
| | | Two birds sit in the grass next to each other. | Bird Sit |
| | | Two black birds are sitting on top of a mountain. | Black Bird. |
| a small boy is eating from a green plate | boy eat | A young boy is enjoying his pizza at the dinner table. | Boy Enjoy |
| | | The little girl is eating lunch and having milk. | Girl Eat |
| | | The woman is eating her meal at the table by herself. | Woman Eat |
| | | An elderly couple is having a small snack in their kitchen. | Couple Have. |
| A brown dog is on the deck of a boat on water. | Brown Dog | A white and black dog laying on top of a yellow boat. | Black Dog |
| | | a brown and black horse some green grass and some houses | Brown Horse. |
| | | The black and white puppy is playing with a small toy. | Dog Play |
| | | A white and black animal lays on a bench that is on grass outdoors. | White Animal |
| a blue bus with a large sign on the side of it. | Blue Bus | A red bus driving down a street in front of a red double decker bus. | Red Bus |
| | | a red car driving down a city road on a cloudy day | Red Car. |
| | | A red bus driving next to an orange and green bus. | Green Bus |
| | | a red double decker bus a regular bus and a tow truck outdoors. | Regular Bus |
| blue bus parked in front of an azure building. | Blue Bus | Two men in suits stand in front of a blue and white semi truck. | Blue Truck |
| A | | a white and black bus with a rainbow colored flag on the front | Black Bus. |
| | | Four friends stand in front of an orange van. | Orange Van |
| | | A large blue RV parked outside a large brick building. | Blue RV |

Table 3: Episode examples constructed by *MetaReVison*'s retrieval modules.

## C Compositional Extracting Rules

After parsing by Stanza, we can extract compositional pairs using the following rules. Compared with Jin et al.' phase extracting rule, MetaReVision extracts more reasonable compositional pairs.

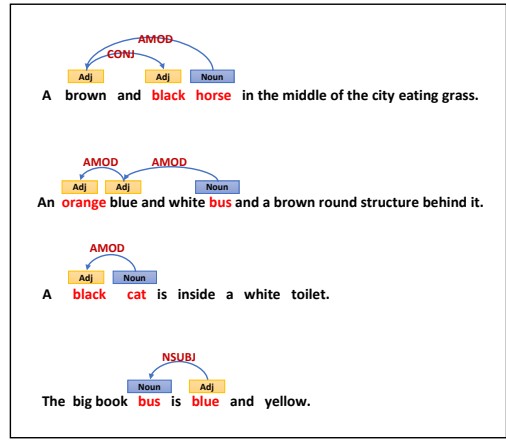

(a) Rules to extract adj-noun pairs.

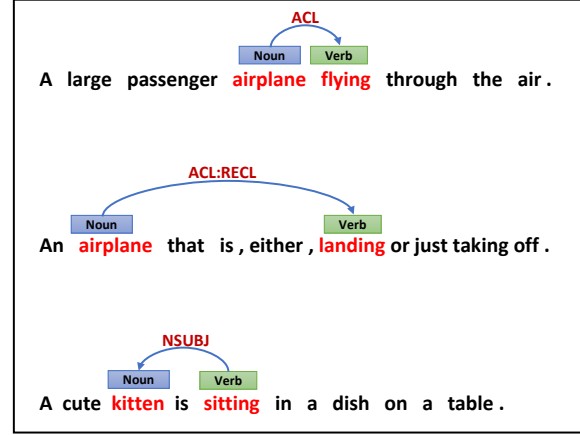

(b) Rules to extract verb-noun pairs.

Figure 8: Extracting rules to Construct CompFlickr and CompCOCO.

# D  *GCCL* Data Statistics

Table 4 shows the statistics of the extracted novel compositional concepts. From the table, we can see that *CompCOCO* has more novel pairs than *CompFlickr*. And *CompCOCO* is a more reliable evaluation for novel compositional learning than And *CompFlickr*

| | MSCOCO | | | | Flickr30K | | | | | |
|---|---|---|---|---|---|---|---|---|---|---|
| | Train Img. | Train Caps. | Test Img. | Test Caps. | Train Img. | Train Caps. | Val Img. | Val Caps. | Test Img. | Test Caps. |
| black bird | 205 | 323 | 122 | 190 | 17 | 24 | 0 | 0 | 2 | 3 |
| small dog | 681 | 1067 | 316 | 481 | 360 | 612 | 11 | 12 | 17 | 33 |
| white boat | 373 | 261 | 196 | 134 | 69 | 85 | 0 | 0 | 3 | 8 |
| big truck | 417 | 601 | 191 | 288 | 28 | 38 | 0 | 0 | 1 | 1 |
| eat horse | 212 | 378 | 106 | 187 | 2 | 2 | 0 | 0 | 0 | 0 |
| stand child | 1288 | 1556 | 577 | 741 | 1048 | 1475 | 38 | 57 | 26 | 36 |
| white horse | 264 | 500 | 151 | 300 | 51 | 100 | 3 | 4 | 4 | 8 |
| big cat | 184 | 216 | 103 | 108 | 0 | 0 | 0 | 0 | 1 | 1 |
| blue bus | 276 | 506 | 143 | 243 | 11 | 16 | 0 | 0 | 0 | 0 |
| small table | 261 | 296 | 134 | 154 | 48 | 54 | 1 | 1 | 1 | 1 |
| hold child | 1328 | 1860 | 664 | 992 | 835 | 1289 | 27 | 37 | 35 | 60 |
| stand bird | 532 | 831 | 260 | 406 | 13 | 24 | 0 | 0 | 0 | 0 |
| brown dog | 613 | 878 | 291 | 430 | 934 | 1838 | 31 | 61 | 29 | 58 |
| small cat | 252 | 325 | 149 | 183 | 2 | 3 | 0 | 0 | 0 | 0 |
| white truck | 262 | 420 | 121 | 175 | 35 | 42 | 2 | 2 | 2 | 2 |
| big plane | 967 | 1345 | 357 | 494 | 5 | 5 | 0 | 0 | 0 | 0 |
| ride woman | 595 | 674 | 300 | 330 | 266 | 537 | 8 | 17 | 9 | 23 |
| fly bird | 245 | 526 | 132 | 283 | 29 | 53 | 0 | 0 | 0 | 0 |
| black cat | 840 | 1760 | 448 | 940 | 15 | 27 | 0 | 0 | 1 | 1 |
| big bird | 215 | 291 | 123 | 169 | 24 | 34 | 0 | 0 | 0 | 0 |
| red bus | 566 | 1212 | 232 | 474 | 11 | 20 | 0 | 0 | 1 | 1 |
| small plane | 481 | 833 | 158 | 279 | 13 | 20 | 0 | 0 | 0 | 0 |
| eat man | 555 | 698 | 250 | 314 | 153 | 272 | 4 | 5 | 5 | 10 |
| lie woman | 301 | 388 | 144 | 194 | 145 | 278 | 1 | 2 | 4 | 8 |

Table 4: Novel Pair Statistics for both *CompCOCO* and *CompFlickr*. We use the same 24 pairs to verify the compositional generalization.