# OpenReview forum: "MetaReVision: Meta-Learning with Retrieval for Visually Grounded Compositional Concept Acquisition"
_EMNLP/2023/Conference — EMNLP 2023 Findings_

### Official Review · Reviewer_6gSR · 2023-07-26

**Soundness:** 3

**Excitement:**

3: Ambivalent: It has merits (e.g., it reports state-of-the-art results, the idea is nice), but there are key weaknesses (e.g., it describes incremental work), and it can significantly benefit from another round of revision. However, I won't object to accepting it if my co-reviewers champion it.

**Paper Topic And Main Contributions:**

This paper defined the Grounded Compositional Concept Learning (GCCL) task, collected two datasets (CompFlickr and CompCOCO) for this task, and proposed a MetaReVision method to address this task, which is based on retrieval and meta-learning.

**Reasons To Accept:**

It is novel to integrate retrieval and meta-learning.

**Reasons To Reject:**

It is not clear what practical applications can be associated with the GCCL task, and the proposed MetaReVision method is too complicated, which is hard to follow.

**Reproducibility:**

2: Would be hard pressed to reproduce the results. The contribution depends on data that are simply not available outside the author's institution or consortium; not enough details are provided.

**Reviewer Confidence:**

2: Willing to defend my evaluation, but it is fairly likely that I missed some details, didn't understand some central points, or can't be sure about the novelty of the work.

---

> ### Author Rebuttal · Authors · 2023-08-29
>
> **It is not clear what practical applications can be associated with the GCCL task and the proposed MetaReVision method is too complicated, which is hard to follow.**
>
> **Ans**: We thank the reviewer for acknowledging the novelty of our approach for the integration of retrieval and meta-learning.
> To the first question on the application of GCCL, we believe this approach can leverage various applications in VL area. The straightforward one is image captioning where we want to include detailed composition explanations of objects and attributes.
> For the reading difficulty, it would be great if the reviewer could provide comments on the unclear parts that we can improve.

---

### Official Review · Reviewer_tj2B · 2023-08-02

**Soundness:** 3

**Excitement:**

3: Ambivalent: It has merits (e.g., it reports state-of-the-art results, the idea is nice), but there are key weaknesses (e.g., it describes incremental work), and it can significantly benefit from another round of revision. However, I won't object to accepting it if my co-reviewers champion it.

**Paper Topic And Main Contributions:**

This paper focuses on systematic generalization within the multi-modal setting, proposes a multi-modal compositional problem, Grounded Compositional Concept Learning (GCCL), and proposes two benchmarks: CompFlickr and CompCOCO, which enrich zero-shot vision-language tasks, from end-task level to token-level. MetaReVision is proposed to solve this problem.

**Reasons To Accept:**

- The task of Grounded Compositional Concept Learning is useful.
- The scale of experiments is decent, and the empirical evaluation is solid.

**Reasons To Reject:**

Minor issues:
- Misspell: the first paragraph in the Introduction, `propose a multi-modal copositional problem`
- Misspell: in the first contributions, `address the domain-shift problem in compositioanl learning...`
- Misspell: the last sentence in Compositional Learning, `compptional problem`
- Misspell: Figure 2 ... `predict the masked conpositional concept conditioned`

Major issues:
- In the contribution of this paper, it is mentioned to deal with the domain-shift problem in compositional learning. Which existing works have this problem, or do all the compositional learning have it? It is not mentioned above, and it is better to explain it earlier.
- Please explain the difference between Compositional Learning and Grounded Compositional Concept Learning in one sentence.
- If the difference between the training data and the test data is too large, will the retriever still work?
- Please briefly explain the difference between the retriever and the attention mechanism in this paper.


**Reproducibility:**

3: Could reproduce the results with some difficulty. The settings of parameters are underspecified or subjectively determined; the training/evaluation data are not widely available.

**Reviewer Confidence:**

3: Pretty sure, but there's a chance I missed something. Although I have a good feel for this area in general, I did not carefully check the paper's details, e.g., the math, experimental design, or novelty.

---

> ### Author Rebuttal · Authors · 2023-08-29
>
> **About minor issues**
>
> **Ans:** Thank you for pointing all these out and we will modify all of them in the new version.
>
> --------
>
>
> **Domain Shift Problem:** In the contribution of this paper, it is mentioned to deal with the domain-shift problem in compositional learning. Which existing works have this problem, or do all the compositional learning have it? It is not mentioned above, and it is better to explain it earlier.
>
>
> **Ans**: Yes, all compositional learning has domain-shift problem, because we have not seen the novel compositions during training time. The statistical distribution of the data in the training set (seen compositions) and the testing set (novel compositions) are significantly different. Compositional learning can be viewed as a special case of zero-shot learning problem and the domain shift problem is commonplace in Zero-Shot Learning. We will mention that earlier in the revised version.
>
> -------
>
>
> **difference between Compositional Learning and Grounded Compositional Concept Learning in one sentence**
>
> **Ans:** GCCL is a specific case of compositional leaning which focus on the compositional learning's systematic aspect and the grounding part. Compsitional learning in the vision and language setting is mostly concerned with grounding that is why we call it grounded compositional concept learning. However, there are compositional learning settings that are defined only based on a single modality like natural langue. For example working on longer length text when trained on shorter text, etc.
>
>
> -----
>
> **If the difference between the training data and the test data is too large, will the retriever still work?**
>
> **Ans:**  The performance will degenerate as shown in Fig. 6. This is caused by the domain-shift problem and the challenge of GCCL.
>
>
> --------
>
> **difference between the retriever and the attention mechanism**
>
> **Ans:** Retriever uses the attention mechanism to calculate the primitive concept's embedding and construct the element concept database.
>
> -----

---

### Official Review · Reviewer_c8Aa · 2023-08-04

**Soundness:** 3

**Excitement:**

3: Ambivalent: It has merits (e.g., it reports state-of-the-art results, the idea is nice), but there are key weaknesses (e.g., it describes incremental work), and it can significantly benefit from another round of revision. However, I won't object to accepting it if my co-reviewers champion it.

**Paper Topic And Main Contributions:**

This paper aims to study models' generalization ability to combine primitive concepts into compositional concepts. For example, see the primitive concepts of "red" and "chair" during training and can understand the novel composition of "red chair" during testing. This paper introduced two datasets based on MSCOCO and Flicker to conduct the compositional learning. The proposed MetaReVision method consists of a retrieval module to retrieve relevant primitive concepts and a meta-learning module to train the model with retrieved data. The experiments show that the MetaReVision method can improve the performance of novel compositional concepts.

**Questions For The Authors:**

Line 406/607, how do you know GPT-4 cannot take multiple images?

**Reasons To Accept:**

1. The task/dataset is designed neatly to study the understanding of compositional concepts.
2. The proposed retrieval-enhanced meta-learning is intuitive and proved effective in novel concept recognition.

**Reasons To Reject:**

1. Stronger VL models.
The models used in this paper are LXMERT and VLBERT, which were proposed 3 or 4 years ago. It's better to evaluate more advanced models like BLIP-2. I agree that the large-scale model may have seen the novel concepts during training, which may muddy the experiments. But the absolute numbers of pair predictions in the paper look very low (10%), making me question the proposed method's effectiveness.

2. Generalized to more complicated compositions.
This task/data only evaluate the composition between one adjective and one object. It could make the paper stronger to consider other types/numbers of compositions. It is also not clear to what's the difference between this paper and previous work on compositional concepts like Winoground.

3. Paper writing.
There are some typos in the paper (see later comments). Some figures can be improved, e.g., in Fig 6, hard to see the points of novel pairs.

**Reproducibility:**

3: Could reproduce the results with some difficulty. The settings of parameters are underspecified or subjectively determined; the training/evaluation data are not widely available.

**Reviewer Confidence:**

4: Quite sure. I tried to check the important points carefully. It's unlikely, though conceivable, that I missed something that should affect my ratings.

**Typos Grammar Style And Presentation Improvements:**

Line 315 reresentive - > representative
Line 380: Val inconsistent with the equation
Line 444: inconsistent symbols in the equation and text
Line 458: memory -> memorize

---

> ### Author Rebuttal · Authors · 2023-08-29
>
> **Stronger VL models.**  The models used in this paper are LXMERT and VLBERT, which were proposed 3 or 4 years ago. It's better to evaluate more advanced models like BLIP-2. I agree that the large-scale model may have seen the novel concepts during training, which may muddy the experiments. But the absolute numbers of pair predictions in the paper look very low (10\%), making me question the proposed method's effectiveness.
>
> **Ans:** Really appreciate for the suggestion. We also consider using SOTA models, like BLIP-2 and MiniGPT, in GCCL. However, we stick to LXMERT and VLBERT based on following considerations:
> - *GCCL problem setting*. IN GCCL, we want to explore VL model's compositional learning ability (recognizing attr-obj and obj-verb pairs), given the contextual visual and textual hints and we formulate GCCL as a *[MASK]* token predicting problem. However, SOTA models, like BLIP-2 and MiniGPT, are generative models which are not suitable in GCCL setting.
> - *Controlled evaluation*. As reviewer mentioned, the compositional evaluation in GCCL is controlled and we only train from scratch using our own training data that excludes the novel compositions to avoid muddy evaluation.
> - *Low performance*. This shows that GCCL is a challenging problem. Although large VL models achieve success in downstream tasks, like VQA or retrieval, they have difficulty in this fine-grained compositional generalization.  MetaReVision propose a possible direction, combining retreval and mata-learning, to enhance compositional learning for GCCL, but far from solving it.
>
>
> -------
>
> **Generalized to more complicated compositions.** This task/data only evaluate the composition between one adjective and one object. It could make the paper stronger to consider other types/numbers of compositions.
> It is also not clear to what's the difference between this paper and previous work on compositional concepts like \textit{Winoground}.
>
> **Ans:** We agree with the reviewer that there are much more complex compositional learning settings. However, as the reviewer kindly acknowledged our contribution, this GCCL problem is still very challenging and worth independent investigation.
> The **Winoground** task is indeed an interesting task designed totally differently. It covers the problem of compositionality, testing does not necessarily considers novel compositions. They have a multichoice setting that include exact same words with different orders. The order changes the compositional meaning. As reviewer suggested, retrieval-based meta-learning could also be applied this Winoground task with carefully designed retriever design and learner design, which is for our future work.
>
>
> --------
>
> **Paper writing**
>
> **Ans:** Thank you for pointing all these out and we will modify all of them in the new version.
>
> ---------
>
> **Question:** Line 406/607, how do you know GPT-4 cannot take multiple images?
>
>
>
> **Ans:** Here, we mean that we can't input multiple images to GPT-4 *in one round* as contextual visual information for GCCL. We hope to get the compositional prediction in one round. But interacting/talking with GPT-4, we can input one image in one round and then several images in several round to prompt GPT-4 for GCCL.  But this is another GCCL setting.
>
>
> -------------------

---

### Official Review · Reviewer_SB8Y · 2023-08-21

**Soundness:** 3

**Excitement:**

3: Ambivalent: It has merits (e.g., it reports state-of-the-art results, the idea is nice), but there are key weaknesses (e.g., it describes incremental work), and it can significantly benefit from another round of revision. However, I won't object to accepting it if my co-reviewers champion it.

**Paper Topic And Main Contributions:**

The paper proposed a novel meta-learning framework with retrieval for the Grounded Compositional Concept Learning (GCCL) problem. Specifically, a retriever is used to help construct the support set with similar concepts as the query item for the meta-learner to learn a generic representation. The paper also proposed two benchmark datasets for GCCL and analyzed the proposed framework on these two benchmarks.

**Questions For The Authors:**

Question A: The verbalizer is only introduced in section 4.3 and briefly explained using a figure (Figure 5, line 461-465) but the concrete implementation remains unclear.

Question B: The implementation of the baselines is not directly inferrable from the name: how does MAML work without the retrieved concepts as the support set?

Question C: What are duplicated concepts (line 532)? Does one concept have multiple vector representations in the element concept DB?

Question D: The experiment section should be extended to include more SOTA models to demonstrate the utility and improvement of the proposed framework.

**Reasons To Accept:**

1. As far as the reviewer's knowledge, the proposed framework is novel and the problem is well-motivated.

**Reasons To Reject:**

1. Implementation of part of the proposed framework is not explained clearly (Question A);
2. Some experimental details are unclear (Question B, C);
3. The work is mainly empirical, but the experiment section is generally weak: only numerical performances are compared with a limited number of models, and the utility of the proposed framework remains in question (Question D);
4. There is very limited documentation of the proposed benchmark datasets.

**Reproducibility:**

3: Could reproduce the results with some difficulty. The settings of parameters are underspecified or subjectively determined; the training/evaluation data are not widely available.

**Reviewer Confidence:**

3: Pretty sure, but there's a chance I missed something. Although I have a good feel for this area in general, I did not carefully check the paper's details, e.g., the math, experimental design, or novelty.

**Typos Grammar Style And Presentation Improvements:**

It is strongly suggested to work on correcting the typos throughout the paper (line 38, 217, 457, 489, 527, etc).

---

> ### Author Rebuttal · Authors · 2023-08-29
>
> **Q_A: About the verbalizer**
>
> **Ans:** The verbalizer constructs a smaller vocabulary based on the retrieved items to limit the predictions to those labels. We will make sure to refer to the verbalizer in the beginning of section 4 and clarify the implementation further in section 4.3. The introduction of verbalizer is to enforce the learner to make prediction considering  both the retrieved items and current parameters, which solve the *memorization* problem in meta-learning.
>
> -------
>
> **Q_B: how does MAML work without the retrieved concepts as the support set?**
>
> **Ans:**  As discussed in Sec. 5.4 (line 525-528), we use the same retriever to construct the episodes. However, during test time, we do not have a support set, we just do normal *[MASK]* prediction using the meta-trained VL learner.  This result difference is used to show the importance of retriever in GCCL. GCCL is a zero-shot problem, the retriever help release the learning burden from VL learner to retriever.
>
> --------
>
>
> **Q_C:** What are duplicated concepts (line 532)? Does one concept have multiple vector representations in the element concept DB?
>
> **Ans**: Yes. One concept could have multiple vector representations [when combined with different attributes/objects]. For example, car could have a different vector representation when modified by 'red' or 'blue'. The encoding is affected by different visual and textual contexts.
>
>
> --------
>
> **Q_D:** Extend to SOTA models.
>
> Really appreciate for the suggestion. We also consider using large pre-trained VL models (if that is what the reviewer means by SOTA), like BLIP-2 and MiniGPT, in GCCL. However, we stick to LXMERT and VLBERT based on following considerations:
>
> - *GCCL problem setting*. IN GCCL, we want to explore VL model's compositional learning ability (recognizing attr-obj and obj-verb pairs), given the contextual visual and textual hints and we formulate GCCL as a [MASK] token predicting problem. However, SOTA models, like BLIP-2 and MiniGPT, are generative models which are not suitable in GCCL setting.
> - *Controlled evaluation*. As reviewer mentioned, the compositional evaluation in GCCL is controlled and we only train from scratch using our own training data that excludes the novel compositions to avoid muddy evaluation.
> - *Low performance*. This shows that GCCL is a challenging problem. Although large VL models achieve success in downstream tasks, like VQA or retrieval, they have difficulty in this fine-grained compositional generalization. MetaReVision propose a possible direction, combining retreval and mata-learning, to enhance compositional learning for GCCL, but far from solving it.
>
> -----
>
> **Typos and Presentation Improvements**
>
> Thank you for pointing the typos out and we will modify all in the revised version.
>
> -----

---

### Meta-Review · Area_Chair_cFtw · 2023-09-19

**Recommendation:** 3

**Metareview:**

This paper defined a new Grounded Compositional Concept Learning (GCCL) task, modified two existing datasets into CompFlickr and CompCOCO for this task, and proposed a method based on retrieval and meta-learning for this task,

All reviewers were in consensus that the paper is sound with sufficient experiments backing up their main contribution of combining meta-learning techniques with retrieval methods.

On the flip side, reviewers were also concerned regarding the practical application of the proposed approach, since the main experiments are conducted on synthetically modified versions of existing image-text datasets. There were also concerns that experiments were tested primarily on relatively old methods, and as a result there were no reviewers who felt particularly excited about the new practical applications of the methods proposed in the paper. I would suggest the authors try to tailor their problem setting so that the more recent (generative) vision language models are compatible, which should lead to a more accurate reflection of the difficulty of the problem and can pave the way for more impactful methods designed for the problem. It would also help to have one key experiment on a truly real-world task (not modified from other datasets) in order to showcase how realistic the task actually is.

---

### Decision · Program_Chairs · 2023-10-07

**Decision:**

Accept-Findings

**Comment:**

This paper defined a new Grounded Compositional Concept Learning (GCCL) task, modified two existing datasets into CompFlickr and CompCOCO for this task, and proposed a method based on retrieval and meta-learning for this task,

All reviewers were in consensus that the paper is sound with sufficient experiments backing up their main contribution of combining meta-learning techniques with retrieval methods.

On the flip side, reviewers were also concerned regarding the practical application of the proposed approach, since the main experiments are conducted on synthetically modified versions of existing image-text datasets. There were also concerns that experiments were tested primarily on relatively old methods, and as a result there were no reviewers who felt particularly excited about the new practical applications of the methods proposed in the paper. I would suggest the authors try to tailor their problem setting so that the more recent (generative) vision language models are compatible, which should lead to a more accurate reflection of the difficulty of the problem and can pave the way for more impactful methods designed for the problem. It would also help to have one key experiment on a truly real-world task (not modified from other datasets) in order to showcase how realistic the task actually is.